

# Unfamiliar face matching with photographs of infants and children

Robin S.S. Kramer[1], Jerrica Mulgrew[2] and Michael G. Reynolds[2]

[1] School of Psychology, University of Lincoln, Lincoln, UK
[2] Department of Psychology, Trent University, Peterborough, ON, Canada

## ABSTRACT

**Background:** Infants and children travel using passports that are typically valid for five years (e.g. Canada, United Kingdom, United States and Australia). These individuals may also need to be identified using images taken from videos and other sources in forensic situations including child exploitation cases. However, few researchers have examined how useful these images are as a means of identification.
**Methods:** We investigated the effectiveness of photo identification for infants and children using a face matching task, where participants were presented with two images simultaneously and asked whether the images depicted the same child or two different children. In Experiment 1, both images showed an infant (<1 year old), whereas in Experiment 2, one image again showed an infant but the second image of the child was taken at 4–5 years of age. In Experiments 3a and 3b, we asked participants to complete shortened versions of both these tasks (selecting the most difficult trials) as well as the short version Glasgow face matching test. Finally, in Experiment 4, we investigated whether information regarding the sex of the infants and children could be accurately perceived from the images.
**Results:** In Experiment 1, we found low levels of performance (72% accuracy) for matching two infant photos. For Experiment 2, performance was lower still (64% accuracy) when infant and child images were presented, given the significant changes in appearance that occur over the first five years of life. In Experiments 3a and 3b, when participants completed both these tasks, as well as a measure of adult face matching ability, we found lowest performance for the two infant tasks, along with mixed evidence of within-person correlations in sensitivities across all three tasks. The use of only same-sex pairings on mismatch trials, in comparison with random pairings, had little effect on performance measures. In Experiment 4, accuracy when judging the sex of infants was at chance levels for one image set and above chance (although still low) for the other set. As expected, participants were able to judge the sex of children (aged 4–5) from their faces.
**Discussion:** Identity matching with infant and child images resulted in low levels of performance, which were significantly worse than for an adult face matching task. Taken together, the results of the experiments presented here provide evidence that child facial photographs are ineffective for use in real-world identification.

Corresponding author
Robin S.S. Kramer,
remarknibor@gmail.com

## INTRODUCTION

Research has repeatedly shown that deciding whether two different face photographs are of the same person, or whether a person standing in front of you is the same person depicted in a photograph, results in rapid and accurate assessments for familiar faces (*Bruce et al., 2001*) and inaccurate assessments for unfamiliar faces (*Bruce et al., 1999*, *2001*; *Kemp, Towell & Pike, 1997*; *Megreya & Burton, 2006*, *2008*). Indeed, a benchmark test of unfamiliar face matching found performance levels of around 90% (*Burton, White & McNeill, 2010*), representing a 'best case' scenario since images were high quality and taken only minutes apart. This detriment with unfamiliar face matching has important implications for real-world professions (e.g. border control situations) and theories of face perception. It is worth noting, for example, that passport officers are no better than the general population on such tasks (*White et al., 2014*).

The focus of the present paper is the accuracy of matching infant identities using faces. Face matching research has concentrated on adult faces, and to our knowledge, there has been little consideration of how difficult this task may be with infants. This is surprising, given the practical implications of validating the identity of an infant. Consider the issue of identifying children in border control situations. It is estimated that up to 400,000 children are trafficked across international borders annually (*U.S. Department of State, 2007*). Furthermore, baby-selling and illegal adoption have been reported in Europe, Africa, Central and South America, Central Asia and East Asia (*UNODC, 2016*). In many countries, infants are required to travel using their own passports as photographic ID in order to combat trafficking, and so it is important to determine the efficacy with which infants can be identified using these images.

The goal of the present paper is to examine whether there is an empirical reason to treat infant face matching as different than adult face matching. Based on related research examining how people *recognise* previously seen faces, we hypothesise that infant face matching will be noticeably harder than adult matching. For instance, we know that viewers show an own-age bias when recognising faces, where people are better at recognising previously seen faces of one's own age group (*Rhodes & Anastasi, 2012*), with adults showing worse recognition of previously seen infants based on their faces (*Chance, Goldstein & Andersen, 1986*). This bias seems to be at least partially based on experience (*Harrison & Hole, 2009*), although the quality of exposure to infant faces may be more important than the quantity (*Yovel et al., 2012*). While researchers have yet to consider the possibility of such a bias when matching identities based on faces, it is likely to be a problem given that infant faces appear more alike than adult faces for adult viewers (based on subjective ratings; *Chance, Goldstein & Andersen, 1986*). For the analogous situation with own-versus other-race faces, evidence has shown that an own-race bias is present in both face recognition (*Meissner & Brigham, 2001*) and matching tasks (*Megreya, White & Burton, 2011*; *Meissner, Susa & Ross, 2013*). This suggests that performance may show the same detriment as with any other group with which we have little experience. In addition, it is possible that children's faces are simply more homogeneous than adult faces. For example, craniofacial shape cues to an identity's sex

are more pronounced after puberty (*Enlow, 1982*), resulting in prepubescent children displaying less between-face variability. If this is the case, we should expect a serious failure regarding the use of infant photos in matching.

If adults have difficulty recognising infants, then this problem may persist when deciding whether two photographs depict the same child. Furthermore, given that there are dramatic changes in facial structure during childhood, comparing photographs of infants of similar ages might even be considered an 'easy' context in relative terms. Many countries that require infants to have their own passports allow them to be used for five years before expiration (e.g. Canada, United Kingdom, United States and Australia). This means that officials may need to compare two images (or an image and a live face) that differ in age by up to five years. Such age gaps likely result in significant difficulties because infants' faces change substantially at a young age (*Chakravarty et al., 2011*; *Farkas et al., 1992*; *Ferrario et al., 1999*), and unfamiliar face matching is closely bound to the visual properties of the particular images (*Hancock, Bruce & Burton, 2000*). Consistent with this possibility, there is evidence that hit rates (correct identification of targets) for adult faces decrease with only 17 months passing between two photographic sittings (*Megreya, Sandford & Burton, 2013*) and that larger age gaps result in worse matching performance (using three images each of four female students; *Seamon, 1982*). Indeed, with an adult face matching test featuring images with an average of only nine months passing between sittings, performance was significantly lower than for the equivalent test where images were taken only minutes apart (*Fysh & Bindemann, 2018*).

Despite the importance of this issue for policy decisions, we were surprised to find only two studies reporting on face matching with images of children, both of which support our supposition that this task is particularly difficult. In a study on machine face recognition, *Yadav et al. (2014)* found poor face matching performance for full-face images of children (60%, where chance level was 50%) when the two photographs depicted an individual once in the age range 0–5 years and again in the range 6–10 years. However, few details about the human task were reported, and the two images presented did not differ by a constant age gap or focus specifically on the change over the first five years of life. A similar conclusion was reported by *White et al. (2015)*. In their task, for trials involving child matching, the recent photograph depicted an individual aged between six and 13 years ($M = 10.0$ years), while their previous photograph was taken an average of 6.2 years earlier. Overall performance on these trials was 39%, with accuracy on adolescent (41%) and adult trials (45%) also very poor, and the latter resulting in statistically better performance in comparison with both child and adolescent trials. Although age was not the focus of this work, these results highlight just how difficult face matching can be with child images, and also provide evidence that this task may be significantly more difficult than adult face matching.

Importantly, neither of these previous studies specifically examined face matching for infants or focussed on the age range considered valid for passports. In the current work, we investigate how difficult face matching is when the images are of infants and young children. Our first experiment explored face matching when both images depicted infants (<1 year old), while our second experiment considered the five-year validity of

child passports by pairing an infant's photograph with one of a child aged 4–5 years old. Experiments 1 and 2 examined performance under optimistic conditions. In these experiments, there was no attempt to make it difficult to detect matches and mismatches in the facial identities presented (e.g. identity pairs were not systematically matched for sex, hair colour, etc.). In contrast, Experiment 3A examined performance when there was a deliberate attempt to make the identities look dissimilar on match trials and similar on mismatch trials, while Experiment 3B extended this further by specifically including only same-sex mismatch pairings. Finally, Experiment 4 addressed this issue of whether the sex of infants and children could be judged accurately from facial photographs, a question which was raised by the first three experiments. Together, we aimed to investigate for the first time how difficult these infant and child face matching tasks may be, and as a result, we hoped to determine the utility of facial images in infant and child passports.

## EXPERIMENT 1

This first experiment examined how accurate people were at deciding whether the infant faces depicted in two photographs belonged to a single individual ('match' condition) or different individuals ('mismatch' condition). Participants were shown pairs of images where both photographs depicted infants in their first year of life and were asked to decide whether these images showed the same infant or two different infants. Comparing two passport-style images mirrors passport replacement and renewal procedures, which are typically carried out online or by post and do not involve 'live matching' to a person. There was no attempt to pair the identities in mismatch trials based upon visual similarity, permitting us to establish an upper-bound for accuracy.

### Method
#### Participants
Thirty students (26 women; age $M$ = 24.90 years, SD = 9.68; 73.33% self-reported ethnicity as White) at Trent University took part in exchange for course credits. All participants in Experiments 1 and 2 provided written informed consent and were verbally debriefed at the end of the experiment. Sample size was based on past research using a face matching paradigm (*Dowsett & Burton, 2015*; *Estudillo & Bindemann, 2014*). Trent University's ethics committee approved all experiments presented here (ref: 22305), which were carried out in accordance with the provisions of the World Medical Association Declaration of Helsinki.

#### Stimuli
Images from the City Infant Faces Database (*Webb, Ayers & Endress, 2018*) were obtained from its creators. These depicted 33 male and 35 female infants in a total of 154 photographs. In most cases, multiple images were available for each infant, with these typically showing a negative, a neutral, and a positive expression (as the original goal of the database was to investigate infant facial expressions).

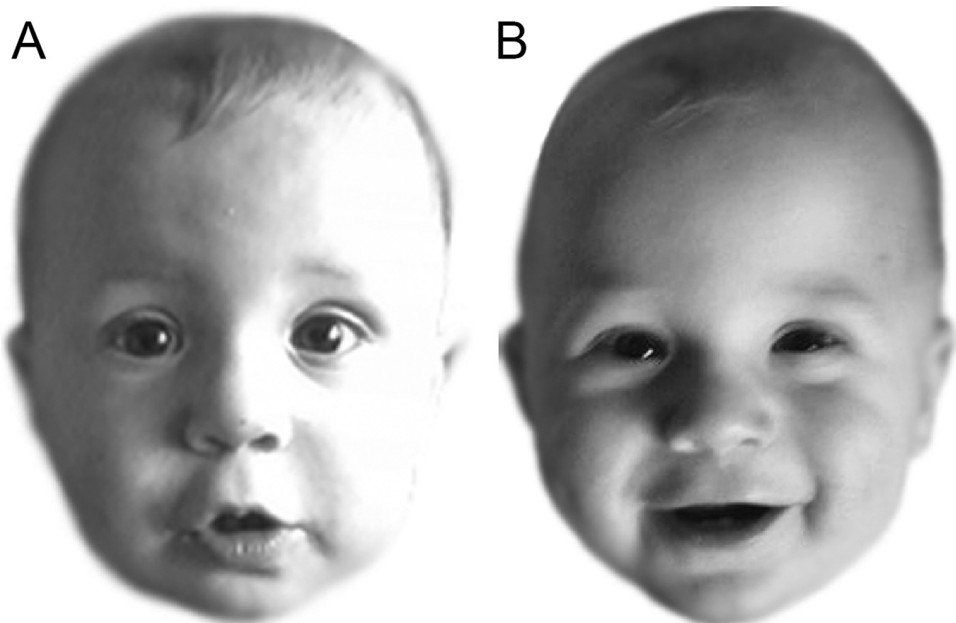

**Figure 1 Example match trial from Experiment 1.** Both images (A) and (B) show the same infant. Figure adapted from *Webb, Ayers & Endress (2018)* (CC BY 4.0).

Each parent was asked to provide multiple images of their infant, and was instructed to take the photographs all at the same time of day and with the infant's head at the same angle in each photograph. No record was kept of when the images were taken, and so these may all have been taken on the same day (at the minimum), or potentially with a few days or weeks between them (at the maximum).

From this original set of 154 images, we considered only infants of White ethnicity where two or more images were available. From this subset, we then excluded images with strong facial expressions, eyes fully closed, strong lighting/shadows, or low resolution. These criteria were used in order to comply with typical governmental guidelines regarding the appearance of standard infant passport photographs. Often, however, guidelines are significantly more relaxed than this (e.g. non-neutral expressions, indirect gaze, and closed eyes can be acceptable; *Her Majesty's (HM) Passport Office, n.d.*) since agencies acknowledge how difficult it can be to capture a controlled infant facial photograph. Finally, where more than two images remained for a given infant, we selected the two displaying expressions closest to neutral. Our final image set comprised 22 male and 19 female infants (age range: 3–11 months), each with two different images that met international standards for passport portraits.

All faces were cropped and shown in greyscale on a white background, and measured approximately 5.5 × 7 cm onscreen (see Fig. 1).

### Procedure

The task comprised 41 match trials (different images of the same infant) and 41 mismatch trials (images of two different infants). The former involved presenting both images of

the infant in the photoset (see above), while the latter were created by pairing one image of every infant (chosen randomly from the two photos available) with an image of a different infant (again, chosen randomly). These infant pairings were created at random for each participant, resulting in every identity appearing four times (two images in a match trial and two in mismatch trials).

Importantly, and as discussed below, identities are not paired at random in real-world contexts, where fraudulent passports would be selected in order to most resemble individuals. Here, the random pairing of identities for mismatch trials meant that it was possible for faces to differ in terms of hair colour, eye colour, and even sex. As such, the present study will likely result in inflated estimates of accuracy.

Participants were tested individually in a computer laboratory. On each trial, two images were presented onscreen, one to the left and one to the right of centre, using custom MATLAB software. Viewing distance was not fixed. The task, which we explained verbally to participants beforehand, was to judge whether the two images were of the same person or two different people. Participants responded using the keyboard, pressing A for 'same' and L for 'different'. These labels were presented at the top of the screen and remained visible throughout the experiment. Trials were self-paced, and no feedback was given at any point during the experiment. The order of the trials was randomised, as was the location of each face (left or right side) within each trial.

Upon completion of the task, demographic information was collected. Participants were also asked if they had had regular contact with infants in the last few years. Unfortunately, very few of our (university student) sample had such experience, and so we were unable to explore this further in our analyses.

## Results

For each participant, we calculated their overall percentage correct. In addition, following other research in this field (*Kramer & Ritchie, 2016*), we investigated signal detection measures. We calculated sensitivity indices (*d'*) and criterion values (*c*) using the following: *Hit*, both images are of the same identity and participants responded 'same'; and *False alarm*, the two images are of different people and participants responded 'same'.

Our results are summarised in Table 1. For this experiment, we found that both percentage correct, $t(29) = 19.98$, $p < 0.001$, Cohen's $d = 3.65$, and $d'$ sensitivity, $t(29) = 16.44$, $p < 0.001$, Cohen's $d = 3.00$, were significantly higher than chance levels. In addition, both hit rate, $t(29) = 9.83$, $p < 0.001$, Cohen's $d = 1.79$, and false alarm rate, $t(29) = 9.93$, $p < 0.001$, Cohen's $d = 1.81$, were significantly better than chance levels. Finally, criterion was not significantly different from zero, $t(29) = 0.48$, $p = 0.632$, Cohen's $d = 0.09$, suggesting no bias in responses.

Table 1 also includes measures of performance for a benchmark test of face matching in order to provide some comparison regarding difficulty. Using the Glasgow face matching test (GFMT; *Burton, White & McNeill, 2010*), researchers presented 168 pairs of passport-style photographs of adult faces and asked participants to decide whether the images were of the same person or two different people. The images were taken approximately 15 min apart but with different cameras. Importantly, on mismatch trials,

**Table 1 A summary of the results for the current experiments and a benchmark face matching task.**

| Source | Stimuli | % Overall | Hit rate | False alarm rate | d' | c |
|---|---|---|---|---|---|---|
| Experiment 1 | 2 infant faces | 72.0 (6.03) | 0.71 (0.12) | 0.27 (0.12) | 1.26 (0.42) | 0.03 (0.35) |
| Experiment 2 | 1 infant + 1 child face | 64.4 (7.71) | 0.64 (0.11) | 0.35 (0.11) | 0.78 (0.45) | 0.02 (0.23) |
| Experiment 3A | 2 infant faces | 56.4 (7.04) | 0.47 (0.16) | 0.34 (0.17) | 0.37 (0.43) | 0.27 (0.44) |
| Experiment 3A | 1 infant + 1 child face | 52.9 (8.62) | 0.54 (0.15) | 0.49 (0.17) | 0.15 (0.48) | −0.04 (0.39) |
| Experiment 3A | 2 adult faces (GFMT short ver.) | 73.3 (12.7) | 0.76 (0.19) | 0.30 (0.20) | 1.52 (0.92) | −0.11 (0.52) |
| Experiment 3B | 2 infant faces | 55.7 (7.09) | 0.44 (0.21) | 0.33 (0.21) | 0.34 (0.42) | 0.36 (0.62) |
| Experiment 3B | 1 infant + 1 child face | 51.5 (7.63) | 0.53 (0.20) | 0.50 (0.20) | 0.08 (0.43) | −0.02 (0.57) |
| Experiment 3B | 2 adult faces (GFMT short ver.) | 70.4 (12.89) | 0.71 (0.23) | 0.30 (0.23) | 1.34 (0.87) | −0.01 (0.69) |
| Burton, White & McNeill (2010) | 2 adult faces (GFMT) | 89.9 (7.3) | 0.92 (0.08) | 0.12 (0.11) | 2.91 (0.83) | −0.09 (0.35) |
| Bobak, Dowsett & Bate (2016) | 2 adult faces (GFMT) | 87.4 (5.26) | 0.91 (0.07) | 0.12 (0.08) | 2.82 (0.73) | −0.11 (0.32) |
| Burton, White & McNeill (2010) | 2 adult faces (GFMT short ver.) | 81.2 (9.4) | 0.80 (0.14) | 0.18 (0.12) | 2.04 (0.84) | 0.06 (0.39) |

**Notes:**
Values presented are $M$ (SD).
GFMT, The Glasgow face matching test.

identities were paired on the basis of similarity, i.e. foil identities were those faces most similar to the target identities. This feature increases the difficulty of the task and therefore establishes the GFMT as a plausible estimate for adult matching accuracy (at least as applied to the specific identities featured). The present study was statistically compared (using $t$-tests with unpooled variances here and below for comparisons between studies) with the GFMT (long version, described here) using the means and standard deviation values reported by *Burton, White & McNeill (2010)*. As Table 1 illustrates, accuracy on the GFMT was higher than our results presented here, $t(38) = 15.19$, $p < 0.001$, Cohen's $d = 2.49$. The same pattern was also found for $d'$ sensitivity, $t(55) = 18.25$, $p < 0.001$, Cohen's $d = 2.06$. However, criterion was not significantly different in comparison with the GFMT, $t(35) = 1.79$, $p = 0.082$, Cohen's $d = 0.34$.

As noted earlier, there was no attempt to pair infants in mismatch trials based on appearance. Indeed, our random pairing of identities on mismatch trials meant that faces with different hair and eye colour were compared, and even male and female faces. As such, the poor performance levels with infant matching reported here may represent an upper estimate in applied situations (e.g. border control), in that pairing identities based upon visual similarity would increase the difficulty of mismatch trials (see Experiment 3). At best, this suggests that border control officers would incorrectly identify 27% (95% CI [23%, 32%]) of the 400,000 infant cases mentioned earlier if they were all presented in 'mismatch' contexts. In order to gain some insight into the extent to which we were overestimating accuracy levels, we elected to compare percentage correct on mismatch trials where the two identities were the same versus different with respect to sex. Interestingly, we found no difference in performance, $t(29) = 0.81$, $p = 0.425$, Cohen's $d = 0.12$. (Although a within-participants comparison, we report Cohen's $d$ using the pooled estimate of the standard deviation as the standardiser here and throughout, more easily allowing for comparisons with other studies irrespective of their designs). This is surprising, given that research has shown that adult participants were able to categorise

the sex of neonates (*Kaminski et al., 2011*; *Porter, Cernoch & Balogh, 1984*; *Round & Deheragoda, 2002*) and 1–24 month old infants (*Tskhay & Rule, 2016*) at levels above chance. Indeed, the sex of infant faces may be perceived automatically (*Tskhay & Rule, 2016*). However, as mentioned earlier, sex characteristics are more evident after puberty and are likely subtle where present in infants. Therefore, either participants were unable to categorise our infant faces by sex or they neglected to use this information when making same/different judgements. We return to this issue in Experiments 3 and 4.

## EXPERIMENT 2

Although Experiment 1 established that matching with infant faces is more difficult than with adult faces, our task was limited to images taken with minimal time passing between photographic sittings. In reality, professionals are required to carry out face matching comparisons with images that were taken up to five years beforehand. Experiment 2 therefore examined face matching performance for photos taken approximately 4–5 years apart. Each image pair showed an infant (less than one year old) paired with a child aged 4–5 years old. Again, the task was to determine whether the two images showed the same individual (match trials) or different individuals (mismatch trials). It was expected that significant aging across images would have detrimental effects on performance. As with Experiment 1, we made no attempt to pair identities in mismatch trials based upon visual similarity. This should make discrimination of match and mismatch trials relatively easy, and permits direct comparison with Experiment 1. As we were unaware of any database containing photos of infants across the age range 0–5 years, we elected to use photos of celebrities' children due to their widespread availability online. This approach resulted in less control over the photographs with regard to pose and facial expression. The use of child images that incorporate more variation than typical passport photos meant that this task was comparable to a border control situation, where a 'live' face is matched to an infant passport image. Similarly, evidence collected in child abuse cases often features more unconstrained images than passports allow.

### Method

#### Participants

Thirty students (26 women; age *M* = 25.00 years, SD = 8.09; 76.67% self-reported ethnicity as White) at Trent University took part in exchange for course credits. There was no overlap between this sample and those who participated in Experiment 1. Sample size was again based on past research using a face matching paradigm (*Dowsett & Burton, 2015*; *Estudillo & Bindemann, 2014*).

#### Stimuli

Images were downloaded from the Internet using Google Image searches for the names of celebrities' children (e.g. Suri Cruise, the daughter of Tom Cruise and Katie Holmes). We chose to collect photographs from this population because images were often available for the same child at different ages due to extensive media coverage. Using the child's

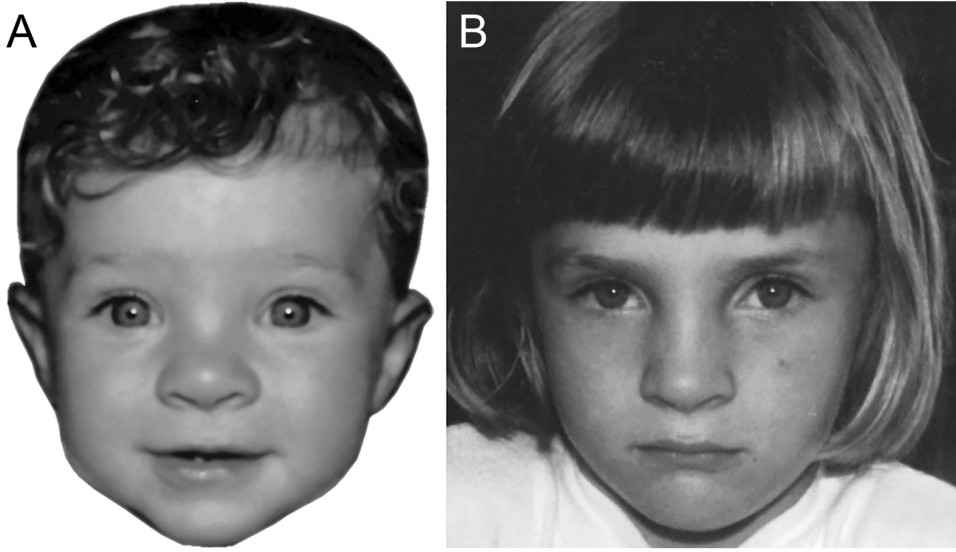

**Figure 2 Images illustrating a match trial in Experiment 2.** The same child is shown at eight months (A) and five years old (B). (Copyright restrictions prevent publication of the original images. Images shown here feature an identity who did not appear in the experiment. This person, now aged 26, has given permission for her images to be reproduced here).

birthdate and the earliest dates when images were posted online, we were able to calculate the approximate age of the child for each image.

For each of 30 children (White ethnicity; 15 female), we collected two photographs—one as an infant (age range: three months to one year old) and the other as a child (age range: 4–5 years old, with the exception of one 6-year-old). Infant images were selected to comply with typical governmental guidelines regarding the appearance of standard infant passport photographs (see Experiment 1). The photographs of the children were taken approximately front-on, with the majority looking directly into the camera and posing with a relatively neutral expression. However, due to the difficulties inherent in collecting these types of images of children, we also accepted slight head turns, as well as some facial expressions (typically, a smile). We felt that some minor deviations from passport photograph guidelines were acceptable for this task and mirrored real-world appearances that might be presented in border control contexts. For this reason, we also chose to leave the child photos uncropped (see Fig. 2). All faces were rotated so that both pupils were aligned to the same transverse plane, and were shown in greyscale on a white background. Images measured approximately 5.5 × 7 cm onscreen.

*Procedure*

The procedure was identical to the one used in Experiment 1, except for the use of different stimuli. Here, 30 match trials and 30 mismatch trials were presented. The former involved presenting both images of the identity (one infant photo and one child photo), while the latter were created by pairing every identity's infant photo with a different identity's child photo (chosen randomly for each participant). In this way, every image appeared twice during the task, once in a match trial and once in a mismatch trial.
Participants were instructed onscreen at the start of the experiment that they would be shown photographs of an infant (one year old or less) and a child (aged 4–5 years), and that their task was to judge whether the two images were of the same child or not.

Given that our images depicted celebrities' children, we asked participants upon completion of the task whether they had recognised any of the identities in the experiment. One participant responded that they had only recognised one identity. We therefore decided not to exclude any data from the subsequent analyses.

Finally, demographic information was collected, and participants were additionally asked if they had had regular contact with infants in the last few years. Unfortunately, as in Experiment 1, very few of our (university student) sample had such experience, and so we were unable to explore this further in our analyses.

## Results

The same performance indicators were examined as in Experiment 1, and can be seen in Table 1. We found that both percentage correct, $t(29) = 10.22$, $p < 0.001$, Cohen's $d = 1.87$, and $d'$ sensitivity, $t(29) = 9.46$, $p < 0.001$, Cohen's $d = 1.73$, were significantly higher than chance levels. In addition, both hit rate, $t(29) = 6.67$, $p < 0.001$, Cohen's $d = 1.22$, and false alarm rate, $t(29) = 7.73$, $p < 0.001$, Cohen's $d = 1.41$, were significantly better than chance levels. Finally, criterion was not significantly different from zero, $t(29) = 0.58$, $p = 0.567$, Cohen's $d = 0.11$, suggesting no bias in responses. For this experiment, we found performance levels that were significantly lower than for Experiment 1: percentage correct, $t(54) = 4.25$, $p < 0.001$, Cohen's $d = 1.10$, and $d'$ sensitivity, $t(57) = 4.27$, $p < 0.001$, Cohen's $d = 1.10$. However, we found no difference between the two experiments with regard to criterion, $t(50) = 0.13$, $p = 0.897$, Cohen's $d = 0.03$.

As with Experiment 1, performance in this study was statistically compared with the long version of the GFMT. As Table 1 illustrates, performance on the GFMT was higher than our percentage correct presented here, $t(34) = 17.36$, $p < 0.001$, Cohen's $d = 3.48$. The same pattern was also found for $d'$ sensitivity, $t(52) = 22.39$, $p < 0.001$, Cohen's $d = 2.65$. In addition, criterion was significantly lower for the GFMT, $t(44) = 2.36$, $p = 0.023$, Cohen's $d = 0.32$.

Our performance level here (64%) was similar to the low levels of accuracy found in previous research investigating matching across age gaps where individuals were depicted once in the age range 0–5 years and again in the range 6–10 years (Yadav et al., 2014). Taken together, it seems clear that comparing an infant's photograph to a real-world image (a little more variation than a typical passport photo) is a highly difficult task, suggesting that border control officers are faced with a significantly error-prone situation.

As in Experiment 1, we compared accuracy on mismatch trials where the two identities were the same versus different with respect to sex. For each participant, we calculated their percentage correct on these two trial types, with a paired samples $t$-test showing no difference in performance, $t(29) = 0.69$, $p = 0.496$, Cohen's $d = 0.17$. Again, we found no evidence that participants were able to categorise our faces by sex and/or use this information when matching. Most likely, this was a failing with the infant faces

(see Experiment 1) since the child photographs (aged 4–5 years) included both hair and some clothing information that made sex categorisation fairly easy.

## EXPERIMENT 3A

This experiment had three objectives. First, we wanted to examine performance under more difficult conditions. In order to make our tasks harder, we followed the same procedure used by *Burton, White & McNeill (2010)* when creating the short version of the GFMT. Difficult versions of our two tasks were created by selecting those trials which demonstrated lowest accuracy in previous participants. By selecting the most difficult trials in our infant and child matching tasks, we mimic realistic cases of fraud where identities are typically selected to be the most likely to fool the authorities. Second, we wanted to compare performance directly with the short version GFMT (*Burton, White & McNeill, 2010*), which provides a more difficult test of adult face matching. Third, by asking each participant to match faces under all three conditions (infant only, infant-child, and adult only using the GFMT), we were able to directly compare performance across stimuli, but also to correlate performance measures in order to determine whether an individual's matching ability in one task was predictive of scores on the other two tasks. Previous research has shown that individuals who perform well on one measure of face matching are typically good at other matching tasks (*Bobak, Dowsett & Bate, 2016*).

### Method
#### *Participants*
A community sample of 114 participants (49 women; age $M = 32.86$ years, SD $= 9.77$; 65.79% self-reported ethnicity as White) were obtained via Amazon's Mechanical Turk (MTurk) in exchange for \$1.50 in payment. There was no overlap between this sample and those who participated in earlier experiments. All participants provided informed consent online and were shown a debriefing screen at the end of the experiment. All participants were unique (due to the nature of the project specifications on MTurk) and so no data were excluded because of repeated participation by the same individuals.

#### *Stimuli*
Images from the short version of the GFMT (*Burton, White & McNeill, 2010*) were used to assess performance for adult face matching under difficult conditions. The task comprised forty pairs of adult male (24) and female faces (16) viewed front on, where half the pairs were match trials (different images of the same person) and half were mismatch trials (different people with a similar appearance). The 40 face pairings were taken from the original GFMT set of 168 pairs (described above) and represent the most difficult trials (based on the performance of 300 participants).

We took similar steps to construct more difficult versions of our two infant matching tasks. In order to select the most difficult 20 match and 20 mismatch trials in each case, we analysed the 'by trial' accuracies for Experiments 1 and 2 and chose the identity pairings that resulted in the lowest performance. This approach was reasonable for match trials because there were 30 observations for each image pair. However, this was more

problematic for mismatch pairs because identities were paired randomly in these experiments. As such, specific pairings occurred infrequently and accuracies were therefore based on small numbers of observations. In all cases, we selected only mismatch trials where the mean accuracy was 0% (i.e. no participant made the correct response), although in some cases, trials were only encountered once previously. As a result, our difficulty manipulation may not have been as powerful as the one used by *Burton, White & McNeill (2010)*. As with the short version of the GFMT, we made no attempt to prevent identities/images appearing more than once (e.g. a particular infant may resemble several others, resulting in their presentation in multiple difficult mismatch trials). As before, we did not restrict ourselves to same-sex pairings in mismatch trials.

All faces were shown in greyscale and measured approximately 6 × 8 cm onscreen.

### Procedure

The experiment was completed online through the Testable website (http://www.testable.org). First, participants were instructed to set their browsing windows to full screen, minimise possible distractions (e.g. TV, phone, etc.), and position themselves at arm's length from the monitor for the duration of the experiment (although viewing distance was not fixed). Next, a screen size calibration took place (adjusting an onscreen bar to match the length of a credit card), consent was obtained, and then demographic information was collected.

On each of the 120 trials (3 tasks × 40 trials), two images were presented onscreen, one to the left and one to the right of centre. The task was to judge whether the two images were of the same person or two different people. Participants responded using the keyboard, pressing A for 'same' and L for 'different'. These labels remained onscreen throughout the experiment. Trials were self-paced, and no feedback was given at any point during the experiment. The trials were blocked by task (infant only, infant-child, and adult only), with the trial order randomised within each task. The task order was also randomised for each participant.

In order to check whether participants were concentrating during the experiment (since this can be a concern for online studies), we included two additional trials that were randomly inserted into the GFMT short version's trial order for each participant. For one trial, a female image from one of the test trials was paired with itself. Because these were two identical images, participants were expected to respond 'same'. For the other trial, a male image from one of the test trials was paired with a (different) female test image. Because these images depicted a man and a woman, participants were expected to respond 'different'.

### Results

The same performance indicators were examined as in Experiments 1 and 2 and can be seen in Table 1. Data from 21 participants were excluded because they responded incorrectly to one or both of the 'checking' trials. Percentage correct scores for the remaining 93 participants' data were analysed using a repeated measures analysis of variance (ANOVA), comparing the three tasks. We found a significant effect of task,

$F(2, 184) = 165.10$, $p < 0.001$, $\eta^2_p = 0.64$, with pairwise comparisons (Dunn–Šidák corrected here and below) revealing that participants performed better on the short version of the GFMT in comparison with the other two tasks (both $ps < 0.001$). In addition, percentage correct scores were significantly lower in the 'infant-child' task in comparison with the 'infant-infant' task ($p = 0.003$).

We also carried out analyses at a more fine-grained scale, considering percentage correct separately on match and mismatch trials. A 3 (Task: infant only, infant-child, adult only) × 2 (Trial Type: match, mismatch) within-subjects ANOVA found a significant main effect of Task, $F(2, 184) = 165.10$, $p < 0.001$, $\eta^2_p = 0.64$, but no main effect of Trial Type, $F(1, 92) = 1.90$, $p = 0.171$, $\eta^2_p = 0.02$. However, these effects were qualified by a significant Task × Trial Type interaction, $F(2, 184) = 34.94$, $p < 0.001$, $\eta^2_p = 0.28$. We therefore considered the simple main effects of Task at each level of Trial Type. These simple main effects were significant for both match, $F(2, 184) = 120.34$, $p < 0.001$, $\eta^2_p = 0.57$, and mismatch trials, $F(2, 184) = 47.06$, $p < 0.001$, $\eta^2_p = 0.34$. Pairwise comparisons showed that, for match trials, accuracies on all three tasks significantly differed from each other (all $ps < 0.001$), in descending order of adult only, infant-child, then infant only. For mismatch trials, accuracies on all three tasks also significantly differed from each other (all $ps < 0.051$), in descending order of adult only, infant only, then infant-child.

An analysis of the $d'$ sensitivities for the three tasks showed the same pattern of results as for percentage correct scores (see Table 1). We found a significant effect of task, $F(2, 184) = 165.87$, $p < 0.001$, $\eta^2_p = 0.64$, with pairwise comparisons revealing that participants showed higher sensitivity on the short version of the GFMT in comparison with the other two tasks (both $ps < 0.001$). In addition, $d'$ sensitivities were significantly lower in the 'infant-child' task in comparison with the 'infant-infant' task ($p = 0.001$).

An analysis of criterion found a significant effect of task, $F(2, 184) = 31.86$, $p < 0.001$, $\eta^2_p = 0.26$. Pairwise comparisons revealed that criterion was significantly higher for the 'infant-infant' task in comparison with the other two tasks (both $ps < 0.001$; see Table 1), with these two tasks not differing from each other ($p = 0.560$).

Although performance was very low in the two infant tasks, we did find that both percentage correct (both $ps < 0.002$) and $d'$ sensitivity (both $ps < 0.003$) remained significantly higher than chance levels on each task. Criterion did not differ from zero in the 'infant-child' task, $t(92) = 1.10$, $p = 0.273$, Cohen's $d = 0.11$, suggesting no bias in responses. However, this measure was significantly above zero in the 'infant-infant' task, $t(92) = 5.97$, $p < 0.001$, Cohen's $d = 0.62$, suggesting a bias towards responding 'different'.

As in Experiments 1 and 2 we compared accuracy on mismatch trials where the two identities were the same versus different with respect to sex. For each participant, we calculated their percentage accuracy on these two trial types, separately for the two infant tasks. Paired samples $t$-tests showed no difference in performance for the 'infant-child' task, $t(92) = 1.10$, $p = 0.275$, Cohen's $d = 0.11$, but a significant difference for the 'infant-infant' task, $t(92) = 4.47$, $p < 0.001$, Cohen's $d = 0.46$. This result suggests that, in contrast with Experiment 1, participants' accuracies were higher on mismatch trials where a male and a female infant were presented together ($M = 68.9\%$) in comparison with two

same-sex infants ($M = 59.6\%$) for this image set. These findings provide mixed support for previous research where infant sex was shown to be accurately judged from facial images (*Tskhay & Rule, 2016*).

In contrast with Experiments 1 and 2, the identity pairings, and hence the trials, for our two infant tasks here were the same for all participants. This allowed us to measure the internal reliability of the two tasks. For each task, we randomly divided the 20 match trials into two sets of ten. We then calculated participants' accuracies for these two sets of trials separately (always using the same two sets, irrespective of the actual order in which they were presented during the task). As a measure of split-half reliability, we correlated these two accuracies across participants, finding significant associations for both the 'infant-infant' task, $r(91) = 0.36$, $p < 0.001$, and 'infant-child' task, $r(91) = 0.31$, $p = 0.002$. Carrying out the same process for mismatch trials, we again found significant associations for both the 'infant-infant' task, $r(91) = 0.56$, $p < 0.001$, and 'infant-child' task, $r(91) = 0.41$, $p < 0.001$. These values were likely lower than for previous tests (e.g. $r = 0.81$ for the GFMT; *Burton, White & McNeill, 2010*) due to the low number of trials in each 'half'. However, it was important to consider the match and mismatch trials separately because previous research has found a dissociation between accuracies on these two trial types (*Megreya & Burton, 2007*).

Finally, we investigated within-person performance across all three tasks. It is well established that face matching ability appears to be a stable trait that generalises across different tasks (*Bobak, Dowsett & Bate, 2016*; *Robertson et al., 2016*), as well as different versions of the same task (e.g. frontal and profile versions of the GFMT; *Kramer & Reynolds, 2018*). For instance, *Bobak, Dowsett & Bate (2016)* reported a correlation of 0.72 between participants' $d'$ values on the GFMT and a second face matching task. Here, we found that $d'$ sensitivities for the short version of the GFMT showed medium-sized correlations with both the 'infant-infant' task, $r(91) = 0.35$, $p = 0.001$, and 'infant-child' task, $r(91) = 0.40$, $p < 0.001$. A similar-sized association was also found between the two infant tasks, $r(91) = 0.28$, $p = 0.007$. Although not as large as the correlation between adult matching tasks previously reported, these results may suggest that an underlying ability with faces supports both adult and infant matching performance. However, it is important to note that $d'$ values were very low and close to chance levels for both infant tasks. With such poor performance, any apparent associations between tasks may simply be due to noise. Therefore, the next experiment will determine whether these within-person correlations can be replicated.

## EXPERIMENT 3B

The results of Experiment 3A suggested that, in contrast with Experiment 1, participants were able to use information regarding the sex of infants in order to perform more accurately on mismatch trials where a male and a female infant were presented together. We therefore decided to rerun Experiment 3A while presenting only same-sex mismatch trials. This would provide us with a full replication of the main result (adult face matching is easier than infant matching tasks) while simulating more real-world contexts, where fraudulent passports would be selected in order to most resemble individuals.

If sex information is available in infant facial photographs then identity pairings would certainly be matched by fraudsters on this dimension. Finally, this experiment will allow us to determine how robust the within-person performance correlations are that were revealed by Experiment 3A.

## Method

### Participants

A community sample of 130 participants (53 women; age $M = 33.18$ years, SD = 9.79; 47.69% self-reported ethnicity as White) were obtained via MTurk in exchange for $1.50 in payment. There was no overlap between this sample and those who participated in earlier experiments. All participants provided informed consent online and were shown a debriefing screen at the end of the experiment. All participants were unique (due to the nature of the project specifications on MTurk) and so no data were excluded because of repeated participation by the same individuals.

### Stimuli

The stimuli used here were identical to those used in Experiment 3A. However, when selecting difficult mismatch trials for the 'infant-infant' and 'infant-child' tasks, we only included same-sex identity pairings.

### Procedure

This was identical to the procedure used in Experiment 3A.

## Results

The same performance indicators were examined as in Experiments 1 and 2, and can be seen in Table 1. Data from 11 participants were excluded because they responded incorrectly to one or both of the 'checking' trials. Percentage correct scores for the remaining 119 participants' data were analysed using a repeated measures ANOVA, comparing the three tasks. We found a significant effect of task, $F(2, 236) = 157.17$, $p < 0.001$, $\eta^2_p = 0.57$, with pairwise comparisons revealing that participants performed better on the short version of the GFMT in comparison with the other two tasks (both $p$s < 0.001). In addition, percentage correct scores were significantly lower in the 'infant-child' task in comparison with the 'infant-infant' task ($p < 0.001$).

We also carried out analyses at a more fine-grained scale, considering percentage correct separately on match and mismatch trials. A 3 (Task: infant only, infant-child, adult only) × 2 (Trial Type: match, mismatch) within-subjects ANOVA found a significant main effect of Task, $F(2, 236) = 157.17$, $p < 0.001$, $\eta^2_p = 0.57$, and of Trial Type, $F(1, 118) = 6.20$, $p = 0.014$, $\eta^2_p = 0.05$. However, these effects were qualified by a significant Task × Trial Type interaction, $F(2, 236) = 27.57$, $p < 0.001$, $\eta^2_p = 0.19$. We therefore considered the simple main effects of Task at each level of Trial Type. These simple main effects were significant for both match, $F(2, 236) = 82.42$, $p < 0.001$, $\eta^2_p = 0.41$, and mismatch trials, $F(2, 236) = 42.98$, $p < 0.001$, $\eta^2_p = 0.27$. Pairwise comparisons showed that, for match trials, accuracies on all three tasks significantly differed from each other

(all $p$s < 0.001), in descending order of adult only, infant-child, then infant only. For mismatch trials, accuracies were lower for the infant-child task in comparison with the other two tasks (both $p$s < 0.001). However, the adult only and infant only tasks did not differ ($p$ = 0.520).

An analysis of the $d'$ sensitivities for the three tasks showed the same pattern of results as for percentage correct scores (see Table 1). We found a significant effect of task, $F(2, 236) = 169.54$, $p < 0.001$, $\eta^2_p = 0.59$, with pairwise comparisons revealing that participants showed higher sensitivity on the short version of the GFMT in comparison with the other two tasks (both $p$s < 0.001). In addition, $d'$ sensitivities were significantly lower in the 'infant-child' task in comparison with the 'infant-infant' task ($p < 0.001$).

An analysis of criterion found a significant effect of task, $F(2, 236) = 24.78$, $p < 0.001$, $\eta^2_p = 0.17$. Pairwise comparisons revealed that criterion was significantly higher for the 'infant-infant' task in comparison with the other two tasks (both $p$s < 0.001; see Table 1), with these two tasks not differing from each other ($p = 0.996$).

Although performance was very low in the two infant tasks, we did find that both percentage correct (both $p$s < 0.030) and $d'$ sensitivity (both $p$s < 0.037) remained significantly higher than chance on each task. Criterion did not differ from zero in the 'infant-child' task, $t(118) = 0.44$, $p = 0.658$, Cohen's $d = 0.04$, suggesting no bias in responses. However, this measure was significantly above zero in the 'infant-infant' task, $t(118) = 6.33$, $p < 0.001$, Cohen's $d = 0.58$, suggesting a bias towards responding 'different'.

Finally, we investigated within-person performance across all three tasks. Here, we found that $d'$ sensitivities for the short version of the GFMT and the 'infant-child' task showed a medium-sized correlation, $r(117) = 0.26$, $p = 0.004$. However, there was no association between the two infant tasks, $r(117) = 0.14$, $p = 0.143$, or the 'infant-infant' task and the GFMT, $r(117) = 0.17$, $p = 0.067$. These results cast doubt on the correlational findings of Experiment 3A, suggesting that any underlying face matching ability may be weaker than was found earlier. As discussed above, the low level of performance for the infant tasks, and the possibility of a floor effect, provide a restricted range with which to investigate any associations, with the risk that any apparent relationships may simply be the result of noise. As such, further work designed specifically to address this issue is needed before firm conclusions can be drawn.

## EXPERIMENT 4

Experiments 1 and 2 provided no evidence that participants were able to perceive or utilise sex information in order to increase performance on mismatch trials where a male and a female identity were presented together. However, the results of Experiment 3A suggested that, at least for 'infant-infant' trials, sex information was indeed beneficial when making same/different judgements. In this final experiment, we therefore decided to investigate whether participants could perceive the sex of infants and children from facial photographs.

## Method

### Participants

A community sample of 40 participants (16 women; age $M = 32.20$ years, SD = 9.36; 70.00% self-reported ethnicity as White) were obtained via MTurk in exchange for $1.50 in payment. There was no overlap between this sample and those who participated in earlier experiments. All participants provided informed consent online and were shown a debriefing screen at the end of the experiment. All participants were unique (due to the nature of the project specifications on MTurk) and so no data were excluded because of repeated participation by the same individuals.

### Stimuli

The stimuli used here were identical to those featured in Experiments 1 and 2. Specifically, we included two images each for 41 identities (Experiment 1) and 30 identities (Experiment 2). This set of 142 images comprised 82 infant faces (Experiment 1), 30 infant faces (Experiment 2), and 30 child faces (Experiment 2).

### Procedure

As with Experiments 3A and 3B, this experiment was completed online through the Testable website. Identical calibration, consent, and debriefing procedures were also used here.

On each of the 142 trials, a single image was presented centrally onscreen. The task was to judge whether the image depicted a boy or a girl. Participants responded using the keyboard, pressing M for 'male' and F for 'female'. These labels remained onscreen throughout the experiment. Trials were self-paced, and no feedback was given at any point during the experiment. The trial order was randomised for each participant and was not blocked by task.

## Results

No data were excluded because the minimum percentage correct score for the children's images (aged 4–5 years old and hence expected to be easier to judge accurately) was 67%. We therefore had no reason to believe that any participants were not paying attention during the experiment.

Percentage correct scores were analysed using a repeated measures ANOVA, comparing the three sets of images (Experiment 1 infants, Experiment 2 infants, Experiment 2 children). We found a significant effect of image set, $F(2, 78) = 271.70$, $p < 0.001$, $\eta^2_p = 0.87$, with pairwise comparisons revealing that participants performed differently across the three sets (all $p$s $< 0.001$). Accuracy was highest for the Experiment 2 children ($M = 82.3\%$, SD = 6.6%), followed by the Experiment 1 infants ($M = 56.7\%$, SD = 5.5%), and then finally the Experiment 2 infants ($M = 50.7\%$, SD = 7.1%).

We found that performance was significantly higher than chance levels for both Experiment 1 infants, $t(39) = 7.74$, $p < 0.001$, Cohen's $d = 1.22$, and Experiment 2 children, $t(39) = 30.72$, $p < 0.001$, Cohen's $d = 4.86$, suggesting that sex information was present in these image sets. However, participants' accuracies were no different from chance for the Experiment 2 infants, $t(39) = 0.60$, $p = 0.555$, Cohen's $d = 0.09$. These

results align with the findings of Experiment 3A, where performance was only higher for different-sex in comparison with same-sex mismatch trials for Experiment 1 images. Taken together, we can conclude only that limited information (accuracy was 57% for Experiment 1 images) regarding infant sex is present in some cases but not others, suggesting the need for further work in this area.

## GENERAL DISCUSSION

Our results provide compelling evidence that matching two images of infants was difficult (72%; Experiment 1), and significantly more so than with two images of adult faces taken from a university population (around 87–90%; *Bobak, Dowsett & Bate, 2016*; *Burton, White & McNeill, 2010*). As we might expect, task performance was significantly lower still when we introduced a five-year age gap between the two images (64%; Experiment 2). Importantly, our estimates of accuracy in these two experiments may even be higher than those found in similar real-world contexts since characteristics such as sex, hair colour, etc. differed on some mismatch trials. For those who make use of fraudulent passports (altered in some way, or simply not their own), the choice of who should be paired with which document/photograph will be driven by facial similarity, which makes the job of spotting mismatches that much harder.

Experiments 3A and 3B addressed how performance changes under more challenging discrimination conditions by examining decisions when presented with more difficult versions of our two tasks. Only the lowest performance trials were included in order to simulate purposeful (rather than random) pairing of infants, as we would predict in real-world fraudulent documents. In addition, Experiment 3B included only same-sex pairings on mismatch trials. Unsurprisingly, performance levels in these experiments were closer to chance (52–56%). This clearly demonstrated that, at its most difficult, using face photographs of infants provided almost no useful information, with accuracies significantly lower than those found with adult faces comparatively selected for difficulty. Importantly, even recent adult matching tasks, specifically constructed to be challenging, showed higher levels of accuracy—66% for the short version of the Kent face matching test (*Fysh & Bindemann, 2018*), 72% for matching with male models (*Dowsett & Burton, 2015*), and 83% for other-race faces (*Kokje, Bindemann & Megreya, 2018*).

We constructed our more difficult task versions through selecting those trials which demonstrated low accuracy in previous participants (mirroring construction of the short version of the GFMT; *Burton, White & McNeill, 2010*). While this does not tell us *why* these particular trials were difficult, we can provide some initial insights through inspection of the image pairings used in Experiments 3A and 3B. First, match trials resulted in poor performance when the image characteristics (e.g. lighting direction, image quality, facial expression) significantly differed across the two photographs. For the 'infant-child' task, changes in hair colour or style were often present, suggesting that participants found it hard to ignore these details even though such differences could be expected for an infant over a five-year period. Second, and as a consequence, difficult mismatch trials presented two images where these superficial characteristics were similar. Previous research has demonstrated that unfamiliar matching relies heavily on the visual

properties of the particular images (*Hancock, Bruce & Burton, 2000*), and evidence suggests an increasing reliance on the internal facial features (eyes, nose, and mouth) as we become more familiar with a face (*Clutterbuck & Johnston, 2002*, *2004*, *2005*), given that the external features (hair, facial outline, etc.) contain less identity information (*Kramer et al., 2018*). Taken together, it is no surprise that image characteristics and external features strongly influenced matching decisions here.

If infant matching suffers from an over-reliance on the external facial features, it may be possible to improve performance on this task through instructing participants to ignore these potentially uninformative sources of information. Indeed, there is some evidence to suggest that performance with matching unfamiliar adult faces can be improved by displaying only the internal features (*Kemp et al., 2016*). However, this advantage was limited to the most difficult trials only and failed to generalise to a card sorting task (*Kramer et al., 2018*). Further research might consider drawing attention towards or away from certain features of the face in order to improve infant matching performance.

While we might consider various methods that could result in performance increases, it may be that even the highest performing humans and machine algorithms will eventually hit a relatively low maximum level. This is because children's faces appear to be more homogeneous than adult faces, displaying lower levels of between-face variability. With less information to distinguish between identities, such images could simply be insufficient for useful identification and matching in real-world situations. Indeed, performance in the current work certainly suggests that any effective method of improving matching will continue to fall short of practical requirements for what is acceptable in terms of accuracy.

We found a substantial drop in performance when pairs of images depicted a five-year age gap (Experiment 2). Research with adult face matching has shown that images taken only minutes apart produced significant levels of error (*Burton, White & McNeill, 2010*), with performance decreasing even further as months passed by between photographic sittings (*Fysh & Bindemann, 2018*; *Megreya, Sandford & Burton, 2013*). Here, we investigated a larger time frame and an infant sample, a combination which was particularly likely to exhibit sizable appearance changes. As such, although important to demonstrate, that matching under these conditions was difficult for our participants came as no surprise. Whether infant face images provide sufficient information for effective identification in real-world scenarios has yet to be determined.

When attempting to compare performance across different types of stimuli, it is important to consider practically how this can best be achieved. There are two separate, yet related issues. One concerns how the comparisons inform the situations that are likely to be encountered in our everyday lives and the second concerns the inferences that can be made about the fundamental cognitive processes that underlie processing of the stimuli. With regards to the former, here we have demonstrated that, on the whole, it is more difficult to match identities using infant faces than adult faces. However, with regard to the latter, we cannot address whether the same cognitive processes are used when matching identities with infant faces and adult faces. If this were the case, then the

performance differences could be explained in terms of the presence/absence of information in the stimuli (e.g. infants were harder to match than adults because sex characteristics were less salient). If infant and adult face matching utilise different cognitive processes (e.g. does emotion play a role in infant face matching?) then they will constitute different tasks and may yield different performance levels, even when the same information is present. One approach might be to construct both types of stimuli using the same identities and pairings but this would require images of each unfamiliar person as an infant, child, and adult, which is logistically problematic. Perhaps more achievable, researchers might consider performance with adult, infant, and child faces that systematically very on a set of characteristics so that the contribution of each feature can be assessed across stimulus types. Future research should consider how to further address this complex issue.

In Experiments 1 and 2, we found that participants were no better on mismatch trials when the two identities were of different sexes. This result appears to contradict previous work showing that infant and neonate faces could be categorised accurately by sex (*Kaminski et al., 2011*; *Tskhay & Rule, 2016*). However, in these studies, accuracy levels were always low, despite being statistically above chance. The results of Experiments 3A and 4 provided additional evidence more in line with past research, suggesting that sex information is present in some image sets, although even in those cases, performance remained close to chance levels. Therefore, participants in the current work may have benefitted to some degree on different-sex trials, but this advantage failed to produce any noticeable gain in accuracy (see Experiments 3A versus 3B in Table 1), perhaps because other, more salient features may have driven judgements.

We found some evidence of within-person correlations in performance across our two infant tasks and the short version of the GFMT. However, these small to medium effects (Experiment 3A) were unlike the large correlations found in previous work when researchers considered two tasks of adult face matching (*Bobak, Dowsett & Bate, 2016*). Indeed, these associations either decreased or were absent when the experiment was repeated (Experiment 3B). It is possible that adult matching employs somewhat different strategies in comparison with infant matching, which might explain why performance associations were notably lower or absent. However, the low correlations can also be explained by the generally low performance on the infant tasks. The low accuracy introduces restricted range issues and also raises the possibility that much of the variability in performance is due to noise. As such, further work is recommended before any conclusions can be drawn regarding the overlap in strategies/abilities across the tasks.

We note that the majority of participants in Experiments 1 and 2 were women. Previous research has shown that women, but not men, demonstrate an own-sex advantage on match trials, and it also seems that women perform better than men on mismatch trials depicting either sex of face (*Megreya, Bindemann & Havard, 2011*). Whether women show higher levels of accuracy with infant and child face matching is currently unknown and future work might consider this question further.

While previous research suggests that experience with infants may result in improved infant face recognition (*Cassia et al., 2009*), we were unable to test the idea

that infant face *matching* would also be easier with increased experience. Unfortunately, the majority of undergraduate university students have minimal experience with infants. We might predict, for example, that nursery school teachers, parents of young children, and midwives, may all show higher levels of performance in comparison with our sample. However, we would still expect lower levels for infant than for adult face matching, even in these populations. Importantly though, evidence suggests that the quality of exposure may be crucial (*Yovel et al., 2012*), with improvement found only when people are required to individuate faces of a particular category. Therefore, further research might consider the possibility of training through individuation of infants/ children in order to explore whether this may increase performance with new identities.

It is interesting to consider how so-called 'super-recognisers' (*Russell, Duchaine & Nakayama, 2009*) might perform on our tasks of infant and child face matching. While these individuals are remarkably good at both recognition and matching with adult faces (*Bobak, Hancock & Bate, 2016*), researchers have yet to determine how well they perform with other populations (e.g. infants or other-race faces). Anecdotally, some White super-recognisers have reported being better with Black faces, although this may be the result of extensive experience/contact with Black criminals and suspects (*Davis, Jansari & Lander, 2013*). If so, we predict that super-recognisers would perform no better than the general population with our current tasks.

## CONCLUSION

Our experiments represent the first focussed investigation of the utility of infant and child facial photographs for use in identification. Performance in both infant-infant and infant-child matching tasks was noticeably lower than with typical adult face matching tests. Despite the randomised pairing of identities, the low levels of accuracy we observed were only minimally aided (if at all) by the availability of sex category information. Taken together, and in combination with previous work (*White et al., 2015*), our results suggest that such low levels of accuracy mean infant and child facial photographs are ineffective for use in real-world identification, and so alternative methods should be considered.

### Funding
The authors received no funding for this work.

### Competing Interests
The authors declare that they have no competing interests.

### Author Contributions
- Robin S.S. Kramer conceived and designed the experiments, performed the experiments, analysed the data, contributed reagents/materials/analysis tools, prepared figures and/or tables, authored or reviewed drafts of the paper, approved the final draft.

- Jerrica Mulgrew conceived and designed the experiments, performed the experiments, analysed the data, contributed reagents/materials/analysis tools, prepared figures and/or tables, authored or reviewed drafts of the paper, approved the final draft.
- Michael G. Reynolds performed the experiments, authored or reviewed drafts of the paper, approved the final draft.

### Human Ethics

The following information was supplied relating to ethical approvals (i.e. approving body and any reference numbers):

Trent University's ethics committee granted approval for the studies (ref: 22305).

### Data Availability

The raw data are provided in a Supplemental File.

### Supplemental Information

Supplemental information for this article can be found online at http://dx.doi.org/10.7717/peerj.5010#supplemental-information.

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
