# Peer review of "Unfamiliar face matching with photographs of infants and children"

_PeerJ, doi:10.7717/peerj.5010_

## Round 0.1 · original submission · Minor Revisions

Thank you for your submission to PeerJ. Two experts in your field have provided reviews of your article, and I have also reviewed it. Both reviewers were complimentary about your studies and both provide detailed and thoughtful feedback. I will not reiterate their comments here, but I do encourage you to respond to each of their questions in turn. Additionally, I have a few points of feedback of my own. In particular, I feel that the descriptions of the three experiments could each benefit from the provision of some additional details to aid with the clarity of their presentation.

Specifically, regarding Exp 1 I note that 86.7% of participants were female. Do you think this sampling bias might affect your results (i.e. is anything known about sex biases in facial recognition)? Please acknowledge this limitation. Additionally, please can you provide more detailed information regarding what instructions you gave participants prior to them starting the task and where and how the task was conducted.

Regarding Exp 2, were the participants the same individuals who completed Exp 1? I assume not, but please state this explicitly. Additionally, as for Exp 1, the majority of participants were female, again, please acknowledge this limitation. Finally, were the 30 children used as the stimuli male and female? If so, what proportion were female?

Regarding Exp 3, in general I found the methods for this experiment a little unclear. For example, were the participants shown both adult-adult pairings (new stimuli) as well as infant-infant pairings (from Exp 1) and infant-child pairings (from Exp 2)? If so, please clarify this and also say what number of pairings came from Exp 1 vs Exp 2 in addition to the novel 40 adult face stimuli. Additionally, what % of the 40 adult face stimuli were female?

In addition to my comments regarding the description of your methods, I also felt that the Discussion was overly long. For example, the first paragraph felt very repetitious from the Introduction and could probably be omitted.

I believe that if you can respond to the reviewers’ comments, in addition to my own, it is likely that I will be able to accept your article for publication in PeerJ, although of course this is not guaranteed.

Reviewer 1 ·

Basic reporting

This paper describes important research currently missing from the research literature and I believe it should be published in PeerJ once the issues listed below are addressed. I commend the authors for writing a very interesting and well thought out paper. It is simple but effective.

Please ensure you report effect sizes throughout. There are some missing.

Lines 207-213 – this section is slightly confusing as you report results of a ‘benchmark test’ – I would move this sentence down to the next paragraph? You then report the t-tests - % correct, sensitivity and criterion. Please additionally report whether hits and false alarms significantly differed from chance – sometimes these effects can be illuminating, and after all you report the descriptives in Table 1.

Line 213 – you then return to the GFMT. Is this the long version of the GFMT? If so, please report trial numbers. And also explain why here you use the long version, later the short version (p 338).
Line 224 and Table 1 – why not report hit and FA rates and run t-tests comparing with criterion?
Line 230 – would it be possible to report confidence intervals when making this passport officer estimation?
Line 309 – also report hits and false alarms t-tests and effect sizes.

Experimental design

Meets all standards

Validity of the findings

Meets all standards.

Additional comments

Line 234 and 321– I think here you should have run a quick follow up study testing why sex of the infants was not informative in assisting ID judgements, as the results suggest there may be something unusual about the images you used. I would ask a few participants to make sex judgements to each infant image. It should not take long, and it would not need many participants but would answer the question as to whether sex information was somehow missing from your images, that was available in the other studies listed in Line 234. It might suggest that with a different set of images of infants, the results for the sex-incongruent results would differ. However, if the editor feels this is not necessary for publication then I am happy for you to leave as is.

·

Basic reporting

The article is clear and well-written. However, I have several suggestions for improving the quality of the manuscript further.

1. The review of the current face matching literature in the introduction is rather sparse and could be more comprehensive. For example, some key studies are missing from the introduction that set the context of the study, such as Burton et al.’s (2010) work with the GFMT showing baseline accuracy rates of 80% as a best-case scenario, and White et al.’s (2014) work with passport officers. Likewise, there is room for discussion of within-person variability in relation to Experiment 2 (see, e.g., Fysh & Bindemann, 2018; Megreya et al., 2013), and how this impacts face matching performance.

2. In the General Discussion (Lines 486-487), the authors write that “children’s faces are more homogenous than adult faces, displaying lower levels of between-face variability.” This is a logical point, but difficult to reconcile with the findings of Experiment 3 – if infant faces are homogenous, then couldn’t we logically expect a response bias in the direction of classifying faces as the same identity, rather than as different?

3. Would it be possible to also provide an example mismatch trial for the Infant-Infant matching task?

4. The link between Experiments 1 and 2 is not explicitly stated, although the rationale is straightforward. A few lines at the end of the discussion for Experiment 1 to introduce the motivation for Experiment 2 would help clarify this.

5. There is little discussion of the findings of Experiment 2. These could be contextualised in line with the current face matching literature (e.g., within-person variability).

Experimental design

The research question is timely and relevant to face matching in practical settings such as passport control. There is clear value in understanding the extent to which child and infant photographs can be accurately matched. However, I find some aspects of the current experimental design to be problematic. These concerns are specified in further detail below.

1. For Experiment 3, the authors note that they did not restrict themselves to same-sex pairings for mismatch trials in the infant-infant and infant-child matching tasks (Line 373). If this is the case, why are differences between same-sex and different-sex pairings not reported as they are in Experiments 1 and 2? In general, this consistent aspect of the design across experiments is problematic, given that this introduces a clear distinction between match (always same-sex) and mismatch trials (sometimes same-sex), and it isn’t clear as to why the researchers chose to construct their mismatch trials in this way.

2. In line with the above point – given that the trial types are so different here, I do not think that it is appropriate to collapse accuracy across match and mismatch trials when analysing performance. Instead, Experiment 3 would be far more informative if the data were analysed via a 2 (trial: match vs. mismatch) x 3 (matching task: infant-infant vs. infant-child vs. GFMT) repeated measures ANOVA.

3. I also have some further concerns about the selection criteria used for stimuli in Experiment 3, namely that items in the new infant-infant and infant-child matching tasks were based on by-item accuracy from Experiments 1 and 2. If my understanding of the first two experiments is correct, then this means that for each match trial, 30 data points were available, but only 1 data point was available per mismatch trial because these were randomly generated for each participant. Is this correct? If so, then these seem to be rather loose criteria for selection of mismatching stimuli to be included in the final test.

4. In Experiment 3, it is reported that 37 participants were recruited, and then in the results it is reported that data from three participants was excluded, with the analysis being based on the remaining 37 participants. This detail should be amended and/or clarified in the ‘Participants’ section of Experiment 3.

Validity of the findings

The findings are interesting and novel. Considered together, this research suggests that it is extremely challenging to match images of two infants (Experiment 1), as well as infant-to-child photographs (Experiment 2), and that these are notably more difficult than matching two (highly optimised) images of adults (Experiment 3). There are some caveats to this work that should be addressed, however, and there is room to extend these findings further. Again, suggestions for improving the current set of results are provided below.

1. The researchers should consider including some reliability measures for the infant-infant and infant-child matching tasks (e.g., Cronbach's alpha or Split-Half Reliability).

2. In general, the sample sizes are rather small across all experiments, and differences between the construction of the tasks used makes it difficult to assess whether the results are actually replicating between Experiments 1 and 2 to Experiment 3. It would be worth re-running Experiment 3 with a greater number of participants to replicate the main findings (i.e. that infant-infant and infant-child matching is more difficult than matching adult faces), as well as clarify the marginal findings (e.g., correlations between tests), which may have been unstable due to the low sample sizes (e.g., Schönbrodt & Perugini, 2013).

3. The researchers may also want to consider examining how accuracy for infant-infant and infant-child matching relates to additional, more difficult tests of face matching (e.g., Kent Face Matching Test, Model Face Matching Test, Good Bad Ugly Face Test, etc.).

4. In Experiment 1, why was accuracy compared to performance in the GFMT as found by Bobak, Dowsett, et al. (2016), as opposed to Burton et al. (2010)? Likewise, why is performance in Experiment 2 also not compared to GFMT levels of accuracy?

Additional comments

In general, this research explores an important set of questions and I would like to see the findings published, following some revisions. I hope that the suggestions that are supplied here will be of some value moving forward with the manuscript.

---

## Round 0.2 · accepted · Accept

Thank you for submitting your revised manuscript to PeerJ. I have reviewed all the edits you made, including the inclusion of new analyses of your existing datasets and the addition of data from two new experiments (3b and 4). I believe your revisions have made throughout addressed my and the reviewers' previous recommendations, resulting in a stronger article. It is my pleasure to recommend it for publication in PeerJ.

#